# The Role of Cytokines Produced via the NLRP3 Inflammasome in Mouse Macrophages Stimulated with Dental Calculus in Osteoclastogenesis

**DOI:** 10.3390/ijms222212434

**Published:** 2021-11-18

**Authors:** Megumi Mae, Mohammad Ibtehaz Alam, Yasunori Yamashita, Yukio Ozaki, Kanako Higuchi, S. M. Ziauddin, Jorge Luis Montenegro Raudales, Eiko Sakai, Takayuki Tsukuba, Atsutoshi Yoshimura

**Affiliations:** 1Department of Periodontology and Endodontology, Graduate School of Biomedical Sciences, Nagasaki University, 1-7-1 Sakamoto, Nagasaki 852-8588, Japan; bb55319213@ms.nagasaki-u.ac.jp (M.M.); bb55319201@ms.nagasaki-u.ac.jp (M.I.A.); yamachanpon@nagasaki-u.ac.jp (Y.Y.); ozaki@nagasaki-u.ac.jp (Y.O.); hig-kanako@nagasaki-u.ac.jp (K.H.); 2National Center for Geriatrics and Gerontology, 7-430 Morioka-cho, Obu City 474-8511, Japan; ziauddin@ncgg.go.jp; 3Department of Oral Anatomy, School of Dentistry, Aichi Gakuin University, 1-100, Kusumoto-cho, Chikusa-ku, Nagoya 464-0851, Japan; jlmonrau@dpc.agu.ac.jp; 4Department of Dental Pharmacology, Graduate School of Biomedical Sciences, Nagasaki University, 1-7-1 Sakamoto, Nagasaki 852-8588, Japan; eiko-s@nagasaki-u.ac.jp (E.S.); tsuta@nagasaki-u.ac.jp (T.T.)

**Keywords:** dental calculus, NLRP3 inflammasome, IL-1β, IL-18, osteoclast

## Abstract

Dental calculus (DC) is a common deposit in periodontitis patients. We have previously shown that DC contains both microbial components and calcium phosphate crystals that induce an osteoclastogenic cytokine IL-1β via the NLRP3 inflammasome in macrophages. In this study, we examined the effects of cytokines produced by mouse macrophages stimulated with DC on osteoclastogenesis. The culture supernatants from wild-type (WT) mouse macrophages stimulated with DC accelerated osteoclastogenesis in RANKL-primed mouse bone marrow macrophages (BMMs), but inhibited osteoclastogenesis in RANKL-primed RAW-D cells. WT, but not NLRP3-deficient, mouse macrophages stimulated with DC produced IL-1β and IL-18 in a dose-dependent manner, indicating the NLRP3 inflammasome-dependent production of IL-1β and IL-18. Both WT and NLRP3-deficient mouse macrophages stimulated with DC produced IL-10, indicating the NLRP3 inflammasome-independent production of IL-10. Recombinant IL-1β accelerated osteoclastogenesis in both RANKL-primed BMMs and RAW-D cells, whereas recombinant IL-18 and IL-10 inhibited osteoclastogenesis. These results indicate that DC induces osteoclastogenic IL-1β in an NLRP3 inflammasome-dependent manner and anti-osteogenic IL-18 and IL-10 dependently and independently of the NLRP3 inflammasome, respectively. DC may promote alveolar bone resorption via IL-1β induction in periodontitis patients, but suppress resorption via IL-18 and IL-10 induction in some circumstances.

## 1. Introduction

Periodontitis is an inflammatory disease that leads to the destruction of periodontal tissue, including alveolar bone [1]. In response to external stimuli, such as dental plaque and dental calculus (DC), various types of leukocytes infiltrate periodontal tissue, and release inflammatory mediators, such as prostaglandins, matrix metalloproteinases, and cytokines, which promote periodontal tissue destruction [2]. Among these mediators, interleukin (IL)-1β has the capacity to trigger potent bone resorption [3], and IL-1β has been detected in the periodontal tissue and gingival crevicular exudates of patients with periodontitis, suggesting its involvement in alveolar bone resorption [4,5].

The production of IL-1β is regulated both transcriptionally and post-transcriptionally [6]. The transcription of pro-IL-1β can be triggered by the binding of Toll-like receptors (TLRs), IL-1 receptors, and tumor necrosis factor (TNF) receptors with their ligands, leading to the nuclear translocation of nuclear factor (NF)-κB [7]. The transcriptional activation of the IL-1β promoter results in the synthesis of a 35 kD pro-IL-1β, which is biologically inactive and remains in the cytosol [8]. For the maturation of IL-1β, pro-IL-1β must be processed by caspase-1 [9]. Assembly of the inflammasome, which consists of nucleotide-binding oligomerization domain leucine-rich repeat containing protein (NLR), apoptosis-associated speck-like protein containing a caspase-recruitment domain (ASC), and the cysteine protease pro-caspase-1, converts pro-caspase-1 to active caspase-1 by autocatalysis; caspase-1, in turn, cleaves pro-IL-1β into its mature IL-1β form [10].

IL-18 and IL-33 are also IL-1 family members that can be cleaved by caspase-1 [11,12]. IL-18 is known to be a proinflammatory cytokine that induces the production of interferon (IFN)-γ in T cells [13], and promotes osteoclastogenesis by upregulating receptor activator of nuclear factor-kappa B ligand (RANKL) production in T cells in synovitis in rheumatoid arthritis [14]. IL-33 binds to type 4 IL-1 receptors and is regarded as a Th2-promoting inflammatory cytokine [15]. The precursor form of IL-33 is already biologically active and is inactivated via cleavage by caspase-1 [11,16]. IL-33 has been reported to inhibit osteoclastogenesis [17,18].

DC is a deposit frequently found in periodontal pockets, and is 70–80% inorganic material [19]. The remaining organic component consists of proteins, leukocytes, and microorganisms [20]. The crystalline components of DC are mainly hydroxyapatite, brushite, tricalcium phosphate, and octacalcium phosphate [19,21]. Recent studies have shown that stimulation of neutrophils and macrophages with DC activates the nucleotide-binding oligomerization domain leucine-rich repeat and pyrin domain containing 3 (NLRP3) inflammasome and produces IL-1β [22]. In addition, IL-18 and IL-33 can be produced by mouse macrophages stimulated with DC via the NLRP3 inflammasome. However, the effects of these cytokines stimulated with DC on osteoclastogenesis have not been investigated.

It is difficult to distinguish the effects of DC from those of dental biofilms because DC is always covered with biofilms that contain highly viable bacteria [23]. However, accumulated evidence has shown that there is a clear association between DC deposition and periodontitis [23,24]. Therefore, the purpose of this study is to investigate the role of cytokines produced by mouse macrophages stimulated with DC via the NLRP3 inflammasome during osteoclastogenesis.

## 2. Results

### 2.1. Effects of Culture Supernatants from Mouse Macrophages Stimulated with Dental Calculus on Osteoclast Formation

We stimulated immortalized wild-type (WT) and NLRP3-deficient mouse macrophages with DC, and examined the effects of culture supernatants on osteoclast formation. When bone marrow macrophages (BMMs) primed with M-CSF and RANKL were incubated with the supernatant from the NLRP3-deficient mouse macrophages stimulated with DC, osteoclast formation was promoted, suggesting the effects of pro-inflammatory cytokines in the supernatants (Figure 1a). The supernatant of WT mouse macrophages stimulated with DC induced higher numbers of osteoclasts than the supernatant from NLRP3-deficient mouse macrophages, suggesting the effect of another pro-inflammatory cytokine (Figure 1b). On the other hand, incubation of RANKL-primed BMMs with the supernatant from unstimulated WT or NLRP3-deficient mouse macrophages did not affect the osteoclastogenesis (Appendix A). In contrast to the results of RANKL-primed BMMs, the supernatants from both WT and NLRP3-deficient mouse macrophages stimulated with DC suppressed osteoclast formation in RANKL-primed RAW-D cells (Figure 1a,b). Incubation of RANKL-primed RAW-D cells with the supernatant from unstimulated WT or NLRP3-deficient mouse macrophages did not affect the osteoclastogenesis. 

In the bone resorption assay, the pit area was proportional to the number of osteoclasts in both RANKL-primed BMMs and RAW-D cells (Figure 2a,b). The supernatant of WT mouse macrophages stimulated with DC induced larger pit areas in RANKL-primed BMMs than the supernatant of NLRP3-deficient mouse macrophages, whereas only a few pit areas were detected in controls. RANKL-primed RAW-D cells formed pit areas, although the supernatants from both WT and NLRP3-deficient mouse macrophages inhibited pit formation. When the bone resorption was analyzed by the release of the fluorescent substrate, the fluorescence intensity of each culture supernatant was proportional to the size of the pit area (Figure 2c).

### 2.2. Cytokine Production by Macrophages Stimulated with DC

Next, we analyzed IL-1β production by immortalized WT and NLRP3-deficient mouse macrophages stimulated with DC using enzyme-linked immunosorbent assay (ELISA). WT mouse macrophages produced IL-1β in a dose-dependent manner, whereas NLRP3-deficient mouse macrophages did not. This indicates that IL-1β was produced via NLRP3 by mouse macrophages stimulated with DC (Figure 3a). Although IL-18 was produced by unstimulated WT mouse macrophages, stimulation with DC enhanced its production (Figure 3b). Stimulation of NLRP3-deficient mouse macrophages with DC resulted in no increase in IL-18 production, indicating that IL-18 production was induced via NLRP3 in mouse macrophages stimulated with DC. IL-33, another IL-1 family member possibly processed by caspase-1 [11,16], was not detected in the supernatants of either WT or NLRP3-deficient mouse macrophages (data not shown). IL-10, an anti-inflammatory cytokine, was produced by both WT and NLRP3-deficient mouse macrophages stimulated with DC in a dose-dependent manner, indicating the production of IL-10 independently of the NLRP3 inflammasome (Figure 3c). Tumor necrosis factor (TNF)-α, a pro-inflammatory cytokine, was also produced by both WT and NLRP3-deficient mouse macrophages stimulated with DC (data not shown).

### 2.3. Effect of Recombinant (r)IL-1 Receptor Antagonist (IL-1ra) on Osteoclastogenesis

IL-1β was detected in the culture supernatant of WT mouse macrophages stimulated with DC. Therefore, we investigated whether inhibition of IL-1β using IL-1ra can inhibit osteoclastogenesis in RANKL-primed BMMs incubated with the culture supernatant from mouse macrophages stimulated with DC. Although incubation with the culture supernatant from WT mouse macrophages stimulated with DC significantly increased the number of tartrate-resistant acid phosphatase (TRAP)-positive multinucleated cells in RANKL-primed BMMs, rIL-1ra suppressed their numbers to the same level as those in RANKL-primed BMMs incubated with the culture supernatant from NLRP3-deficient mouse macrophages stimulated with DC (Figure 4a,b). IL-1ra did not suppress the number of TRAP-positive multinucleated cells in RANKL-primed BMMs incubated with the culture supernatant from NLRP3-deficient mouse macrophages stimulated with DC.

### 2.4. Effects of IL-1β, IL-18, and IL-10 on Osteoclast Formation

We examined the effect of each cytokine detected in the supernatants of mouse macrophages stimulated with DC on osteoclast formation.

Incubation of RANKL-primed BMMs with rIL-1β increased the number of multinucleated TRAP-positive cells in a dose-dependent manner (Figure 5a,b). Incubation of RANKL-primed RAW-D cells with rIL-1β also increased the number of multinucleated TRAP-positive cells, although the effect was less prominent. IL-18 is classified as a pro-inflammatory cytokine; however, previous studies have shown that it inhibits osteoclast formation by inducing osteoclast precursor cell apoptosis [25]. Incubation of RANKL-primed BMMs with rIL-18 significantly reduced the number of multinucleated TRAP-positive cells (Figure 6a,b). Incubation of RANKL-primed RAW-D cells with rIL-18 also reduced the number of multinucleated TRAP-positive cells, although the effect was less prominent. IL-10 is an anti-inflammatory cytokine that inhibits osteoclast formation [26]. Incubation of RANKL-primed BMMs with rIL-10 reduced the number of multinucleated TRAP-positive cells, and incubation of RANKL-primed RAW-D cells with rIL-10 reduced the number of multinucleated TRAP-positive cells even more effectively (Figure 6c,d). Because the inhibitory effect of rIL-10 in RAW-D cells was considerably stronger than that in RANKL-primed BMMs, we analyzed the expression of IL-10 receptors in these cells. The expression of both IL-10 receptor α and β genes (*IL10RA* and *IL10RB*, respectively) was significantly downregulated after the priming with RANKL in RAW-D cells, but not in BMMs, consistent with higher sensitivity of RANKL-primed RAW-D cells to IL-10 (Figure 6e).

### 2.5. Effect of Exosomes Isolated from the Culture Supernatant of Mouse Macrophages on Osteoclast Formation

Exosomes are approximately 100 nm secreted vesicles that contain many signaling molecules, such as microRNAs (miRNAs), messenger RNAs, and proteins [27]. Exosomes released from the host cell surface can fuse with the plasma membranes of recipient cells and deliver their contents into the cytoplasm. The components delivered by exosomes to recipient cells result in the alteration of biological response. It has been reported that the exosomes secreted by periodontal ligament cells suppressed IL-1β production through the inhibition of the NF-κB signaling pathway in macrophages [28]. Therefore, we investigated how exosomes from the supernatants of mouse macrophages stimulated with DC affect osteoclast formation in BMMs and RAW-D cells.

Incubation of BMMs with exosomes from the culture supernatant of WT or NLRP3-deficient mouse macrophages stimulated with DC promoted osteoclast formation to a significant extent (Figure 7a,b). The number of TRAP-positive multinucleated cells after incubation with exosomes from the culture supernatant of WT mouse macrophages was significantly larger than that induced by the exosomes from the culture supernatant of NLRP3-deficient mouse macrophages. In contrast, incubation of RAW-D cells with exosomes from the culture supernatant of WT mouse macrophages stimulated with DC significantly inhibited osteoclast formation. The number of TRAP-positive multinucleated cells incubated with exosomes from the culture supernatant of NLRP3-deficient mouse macrophages stimulated with DC was smaller than the control, but not significant.

## 3. Discussion

The purpose of this study was to investigate the effect of cytokines produced by mouse macrophages stimulated with DC on osteoclastogenesis. The IL-1 family cytokines, IL-1β and IL-18, were produced in the culture supernatant of WT mouse macrophages stimulated with DC in a dose-dependent manner, whereas the production of IL-1β and IL-18 did not increase in the culture supernatant of NLRP3-deficient mouse macrophages, suggesting that the production of IL-1β and IL-18 is dependent on the NLRP3 inflammasome. Another IL-1 family cytokine, IL-33, which can be processed by this inflammasome [11], was not detected in the culture supernatant of mouse macrophages stimulated with DC. In contrast, IL-10 was induced by DC in a dose-dependent manner in both WT and NLRP3-deficient mouse macrophages, suggesting that the production of IL-10 is independent of the NLRP3 inflammasome. The culture supernatant from WT mouse macrophages containing IL-1β, IL-18, and IL-10 promoted osteoclastogenesis in RANKL-primed BMMs, however, it inhibited osteoclastogenesis in RANKL-primed RAW-D cells, indicating that this culture supernatant exerted osteoclastogenic and possible anti-osteoclastogenic effects.

The culture supernatant from WT mouse macrophages stimulated with DC contained IL-1β, and significantly enhanced RANKL-primed BMM osteoclastogenesis. rIL-1β also promoted osteoclastogenesis in both RANKL-primed BMMs and RAW-D cells. This is consistent with previous reports that show that purified IL-1 is a bone resorption molecule involved in chronic inflammatory diseases such as rheumatoid arthritis and periodontitis [3,29], and that rIL-1β promotes bone resorption in vitro [30]. When IL-1ra, an antagonist of the IL-1 receptor, was added to the culture supernatant of WT mouse macrophages, the number of TRAP-positive cells decreased to the level found in response to the culture supernatant from NLRP3-deficient mouse macrophages. All of these findings indicate that IL-1β in the culture supernatant of WT mouse macrophages stimulated with DC strongly promotes osteoclastogenesis in RANKL-primed BMMs. These results suggest that DC may play a clinically important role in alveolar bone resorption. In other words, the leukocytes that migrate into periodontal pockets may incorporate minute DC particles and produce IL-1β via the NLRP3 inflammasome. Although DC precipitates on the tooth surfaces and hardly penetrate the epithelium, IL-1β induced by DC may spread to the vicinity of alveolar bone and promote bone resorption around the calculus deposition sites. From the clinical point of view, application of NLRP3 inflammasome inhibitors, such as MCC950 and glyburide, may be useful for prevention of this alveolar bone resorption.

On the other hand, the culture supernatant from WT mouse macrophages stimulated with DC inhibited osteoclast formation in RANKL-primed RAW-D cells. The cytokines that inhibit osteoclastogenesis are considered to be produced in an NLRP3-independent manner, because the inhibitory effect was also observed in the culture supernatant of NLRP3-deficient mouse macrophages. Although IL-18 production was enhanced by stimulation with DC, it was also produced in unstimulated WT and NLRP3-deficient mouse macrophages. IL-18 is known to be a pro-inflammatory cytokine that promotes osteoclastogenesis by upregulating the expression of RANKL [14], but rIL-18 was found to inhibit osteoclast formation in RANKL-primed RAW-D cells in the present study. This is consistent with a previous report showing that IL-18 inhibits osteoclastogenesis by inducing osteoclast progenitor cell apoptosis [25]. Therefore, IL-18 in the culture supernatant of macrophages may contribute to the inhibition of osteoclast formation in RANKL-primed RAW-D cells. IL-10 has been reported to inhibit osteoclastogenesis by suppressing NFATc1 expression and nuclear translocation [26]. In the present study, rIL-10 inhibited osteoclast formation, and the inhibitory effect was stronger in RANKL-primed RAW-D cells than that in RANKL-primed BMMs, suggesting that IL-10 may contribute to the inhibition of osteoclast formation in RANKL-primed RAW-D cells. Exosomes are extracellular vesicles of about 100 nm in size that have attracted much attention in recent years and play a role in intercellular networks [27]. Exosomes contain cytokines and miRNA, along with other proteins and molecules. It has been reported that periodontal ligament cells stimulated with cyclic stretch secrete the exosomes and that exosomes suppress IL-1β production by inhibiting the NF-κB signaling pathway [28]. In this study, exosomes suppressed osteoclast formation in RAW-D cells. Therefore, exosomes in the culture supernatant of mouse macrophages may contribute to osteoclastogenesis inhibition in RANKL-primed RAW-D cells. These results suggest that IL-18, IL-10, and exosomes induced by DC independent of the NLRP3 inflammasome may inhibit alveolar bone resorption around calculus deposition sites.

The culture supernatant of WT mouse macrophages stimulated with DC promoted osteoclast formation in RANKL-primed BMMs, but inhibited osteoclast formation in RANKL-primed RAW-D cells. These opposite effects on osteoclastogenesis could be explained by differences in the sensitivity to cytokines of the osteoclast progenitors. IL-1β promoted osteoclast formation in RANKL-primed BMMs to a greater extent than in RANKL-primed RAW-D cells. Conversely, IL-10 inhibited osteoclast formation in RANKL-primed RAW-D cells to a greater extent than in RANKL-primed BMMs. The different sensitivities of these osteoclast progenitors may reflect differences in the differentiation stage or the purity of the cell populations. BMMs are primary cells freshly isolated from bone marrow containing heterogeneous populations that may indirectly regulate osteoclastogenesis, whereas RAW-D cells were cloned from RAW264.7 cells having homogenous nature of osteoclast precursor populations [31,32]. The RAW264.7 cell line derives from the lymphoma of a male BALB/c mouse infected with Abelson leukemia virus (v-Abl) [33,34]. Previous studies have shown that v-Abl is expressed in RAW-264.7 cells, and enables the cells to proliferate in the absence of macrophage colony-stimulating factor (M-CSF) [35]. Furthermore, the cell line is known to change its characteristics with the number of passages, suggesting the difference between the gene expression profiles of BMMs and RAW-D cells [36]. However, the culture supernatant from mouse macrophages stimulated with DC contains various biologically active molecules, such as cytokines and exosomes. These molecules may interact with each other and affect osteoclastogenesis in an elaborate way. In this study, we were unable to clarify the exact reason why the culture supernatant from mouse macrophages stimulated with DC exhibited different effects on RANKL-primed BMMs and RAW-D cells, but it is likely related to the different differentiation stages of the osteoclast progenitor cells, and a similar range of differentiation stages exists in the human body. It will be necessary to clarify what conditions make osteoclast progenitors sensitive to osteoclastogenic or anti-osteoclastogenic cytokines in future studies.

This study demonstrated that mouse macrophages stimulated with DC produce osteoclastogenic and possible anti-osteoclastogenic cytokines. As osteoclast formation in freshly isolated BMMs was promoted by the culture supernatant of WT mouse macrophages stimulated with DC, the promotion of bone resorption via IL-1β is likely to be the main effect of the induced cytokines. However, IL-18, IL-10, and exosomes induced by DC inhibited osteoclast formation in RAW-D cells; this anti-osteogenic effect may protect against excessive bone resorption as a type of negative feedback. It is necessary to investigate whether DC causes alveolar bone resorption in vivo using animal models. In addition, the anatomical relationship between DC and the levels of these cytokines in periodontal tissue needs to be more precisely defined. Future studies will reveal the exact role of DC in alveolar bone resorption, and give fresh insight into the pathogenesis of periodontitis.

## 4. Materials and Methods

### 4.1. Reagents

Minimum Essential Media alpha (MEMα), Dulbecco’s phosphate-buffered saline (D-PBS) and recombinant mouse IL-10 were purchased from Fujifilm Wako Pure Chemical (Osaka, Japan). Fetal bovine serum (FBS) was purchased from Hyclone (Logan, UT, USA). Total Exosome Isolation Reagent and penicillin-streptomycin were purchased from Thermo Fisher Scientific (Waltham, MA, USA). Recombinant mouse IL-1β was purchased from Tomy Digital Biology (Tokyo, Japan). ELISA kits for mouse IL-1β, IL-18 and IL-10 (DuoSet, R&D Systems, Minneapolis, MN, USA ), recombinant mouse M-CSF, and recombinant mouse RANKL were purchased from R&D Systems (Minneapolis, MN, USA). TRAP staining kit was purchased from Sigma-Aldrich (St. Louis, MO, USA). Bone resorption assay kit 48 was purchased from Iwai Chemicals Company (Tokyo, Japan). Recombinant mouse IL-1ra was purchased from Biovision (Milpitas, CA, USA).

### 4.2. Preparation of DC

DC was obtained from a chronic periodontitis patient who visited Nagasaki University Hospital. The DC was pulverized, treated overnight with 10% sodium hypochlorite to remove the organic components, and washed 10 times with distilled water. The calculus was then filtered using a 48 μm nylon mesh and autoclaved. The calculus sample thus prepared was weighed, adjusted to the appropriate concentrations, and vigorously vortexed before use in assessing cell stimulation [22].

### 4.3. Cell Cultures

Immortalized bone marrow macrophages from WT (C57BL/6) and NLRP3-deficient mice were generated with a J2 recombinant retrovirus (carrying v-*myc* and v-*raf* (*mil*) oncogenes), and provided by Dr. Eicke Latz at the Institute of Innate Immunity, University Hospital Bonn, Germany. These cells were cultured in MEMα supplemented with 10% FBS, 100 U/mL penicillin, and 100 μg/mL streptomycin.

Murine monocytic RAW-D cells were kindly provided by Prof. Kukita (Kyushu University, Fukuoka, Japan) and cultured in MEMα containing 10% FBS with RANKL (20 ng/mL).

### 4.4. ELISA

Mouse macrophages from WT and NLRP3-deficient mice (1.5 × 10^5^ cells/well) were seeded in 96-well plates and stimulated with DC for 8 h. The culture supernatants were harvested and the concentrations of IL-1β, IL-18, and IL-10 were quantified by ELISA, according to the manufacturer’s protocol [22].

### 4.5. Isolation of BMMs

Bone marrow was extracted from the femur and tibiae of six-week-old male BALB/c mice and flushed with MEMα using a syringe. Bone marrow cells were collected by centrifugation, treated with erythrocyte lysis buffer for 10 min at room temperature, washed with D-PBS, and incubated in MEMα supplemented with 10% FBS, 100 U/mL penicillin, and 100 μg/mL streptomycin for 12 h. Then, non-adherent cells were collected and incubated in 10 cm dishes for 36 h in MEMα containing M-CSF (30 ng/mL), 10% FBS, 100 U/mL penicillin, and 100 μg/mL streptomycin. The cells that adhered to the dishes were collected and used as BMMs.

### 4.6. Osteoclastogenesis

BMMs (1 × 10^4^ cells/well) were seeded in 96-well plates and cultured in MEMα supplemented with 10% FBS, 100 U/mL penicillin, and 100 μg/mL streptomycin with M-CSF (30 ng/mL) and RANKL (20 ng/mL) for 48 h. Then, these cells were incubated for 24 h with the same concentration of RANKL in combination with two-fold diluted culture supernatant from WT or NLRP3^−/−^ mouse macrophages stimulated with 500 μg/mL DC. To examine the effects of cytokines, the cells were incubated with 10 μg/mL of mouse rIL-1ra, the indicated concentration of mouse rIL-1β, rIL-18, rIL-10, or the exosomes isolated from mouse macrophage culture supernatant for a final culture period of 24 h.

RAW-D cells (5 × 10^2^ cells/well) were seeded in 96-well plates and incubated overnight. These were cultured with 20 ng/mL RANKL for 48 h. These cells were then incubated for 48 h with the same concentration of RANKL in combination with two-fold diluted culture supernatant from WT or NLRP3^−/−^ mouse macrophages stimulated with 500 μg/mL DC. To examine the effects of cytokines, the cells were incubated with the indicated concentration of mouse rIL-1β, rIL-18, rIL-10, or the exosomes isolated from mouse macrophage culture supernatant for a final culture period of 48 h.

Then, BMMs and RAW-D cells were subjected to TRAP staining. The cells were fixed in 4% paraformaldehyde for 30 min, treated with 0.4% Triton X-100 in PBS at room temperature for 5 min, treated with TRAP staining solution for 20 min at 37 °C and rinsed three times with distilled water. TRAP-positive cells with three or more nuclei were counted as osteoclasts.

### 4.7. Bone Resorption Assay

BMMs (5 × 10^4^ cells/well) were seeded in 48-well plates and cultured in MEMα supplemented with 10% FBS, 100 U/mL penicillin, 100 μg/mL streptomycin containing M-CSF (30 ng/mL), and RANKL (20 ng/mL), but no phenol red for 48 h. Then, these cells were incubated for 72 h with the same concentration of M-CSF and RANKL in combination with two-fold diluted culture supernatant from WT or NLRP3^−/−^ mouse macrophages.

RAW-D cells (5 × 10^4^ cells/well) were seeded in 48-well plates and cultured in MEMα supplemented with 10% FBS, 100 U/mL penicillin, and 100 μg/mL streptomycin containing RANKL (20 ng/mL), but no phenol red for 48 h. Then, these cells were incubated for 72 h with the same concentration of RANKL in combination with two-fold diluted culture supernatant from WT or NLRP3^−/−^ mouse macrophages.

A bone resorption pit assay was performed according to the manufacturer’s instructions. In brief, the culture supernatants from BMM and RAW-D cells were transferred to a black plate. Bone resorption assay buffer was added and the fluorescence was measured. After removing the supernatant, the calcium-coated plates were treated with 5% sodium hypochlorite to remove the cells. Finally, the pit areas were measured using ImageJ image analysis software (version 1.52a; http://imagej.nih.gov/ij/, accessed on 30 September 2021).

### 4.8. qRT- PCR Analysis

For analysis of IL-10 receptor α (*IL-10RA*) and β (*IL-10R**B*) mRNA levels, BMMs (5 × 10^5^ cells/mL) were seeded in 6-well plates and stimulated with 30 ng/mL M-CSF and 20 ng/mL RANKL for 48 h. RAW-D cells (5 × 10^5^ cells/mL) were seeded in 6-well plates and stimulated with 20 ng/mL RANKL for 48 h. Total RNA was extracted from unstimulated and stimulated cells using the RNeasy Mini Kit (Qiagen, Hilden, Germany) with on-column DNase treatment according to the manufacturer’s instructions. For each sample, 2 μg of total RNA was converted to first strand cDNA, using avian myeloblastosis virus reverse transcriptase (Promega, Madison, WI, USA) at 25 °C for 10 min, followed by 50 min at 42 °C, and then 15 min at 70 °C, using a Takara PCR thermal cycler (Takara Bio, Otsu, Japan). The cDNA was treated with RNase H and purified using the QIAprep Spin Miniprep Kit (Qiagen). The primer sequences used were as follows: *IL-RA* forward, 5′-TCTCCAGGGCAGCCTAAGTA-3′ and reverse, 5′-CTGCAGGTGTACCCCAAGTT-3′; *IL-10RB* forward, 5′-CCAACGAAGAAGCCATAGACA-3′ and reverse, 5′-TCAGGATGACACTCTTTCAG-3′; GAPDH forward, 5′-GGAGGAACCTGCCAAGTATG-3′, and reverse, 5′-TGGGAGTTGCTGTTGAAGTC-3′. Comparative quantification of *IL-10RA* and *IL-10RB* was completed with SYBR Premix Ex Taq using the Mx3000 P qPCR System (Agilent Technologies, Santa Clara, CA, USA). The amplification conditions were: 95 °C for 10 s, followed by 40 cycles of 95 °C for 5 s, 58 °C *(IL-RA*) or 56 °C (*IL-10RB*) for 20 s, and a final cycle of 95 °C for 1 min, 55 °C for 30 s, 95 °C for 30 s. A melting curve analysis was used to confirm that the proper PCR products were amplified in all samples. The relative expression ratio of *IL-10RA* and *IL-10RB* mRNA was calculated based on PCR efficiency and the threshold cycle difference for the test sample (RANKL stimulated cells) versus a calibrator (unstimulated cells). Target gene expression was normalized using *GAPDH* gene expression. The mRNA level of the calibrator was set to 1.

### 4.9. Exosome Isolation

Immortalized mouse macrophages (1.5 × 10^5^ cells/well) were seeded in 96-well plates and stimulated with DC for 8 h. Then, the supernatants were collected and centrifuged at 2000× g for 30 min to remove cells and debris [37]. The exosomes were isolated by removing the endoplasmic reticulum using total exosome isolation reagent and centrifugation, and exosomes were resuspended in D-PBS. When BMMs and RAW-D cells were stimulated with exosomes, the amount of purified exosomes added to the osteoclast precursor cells per well was equivalent to the amount of exosomes included in the culture supernatant from macrophages stimulated with DC.

### 4.10. Statistical Analysis

The statistical differences among the groups were assessed using a one-factor analysis of variance (ANOVA) with the Tukey–Kramer test. The differences between two groups were analyzed using *t*-tests. All calculations were conducted using Stat Mate V (ATMS, Tokyo, Japan).

## Figures and Tables

**Figure 1 ijms-22-12434-f001:**
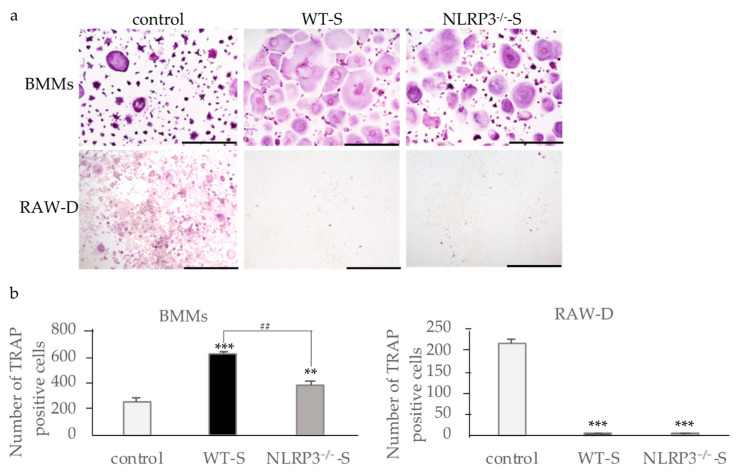
Effects of culture supernatants from mouse macrophages stimulated with dental calculus (DC) on osteoclastogenesis. BMMs were incubated with 30 ng/mL M-CSF and 20 ng/mL RANKL for 48 h. Then, the cells were incubated for 24 h with the same concentration of M-CSF, RANKL, and the culture supernatant from WT or NLRP3^−/−^ mouse macrophages stimulated with DC. RAW-D cells were incubated with 20 ng/mL RANKL for 48 h. The cells were then incubated for 48 h with the same concentration of RANKL and the culture supernatant from WT or NLRP3^−/−^ mouse macrophages stimulated with DC. These cells were subjected to TRAP staining (**a**), and TRAP-positive cells with more than three nuclei were counted (**b**). Scale bar = 500 μm. The differences between the groups were analyzed by one-way ANOVA followed by a Tukey test for multiple comparisons. ** *p* < 0.01 *** *p* < 0.001 compared with the control. ^##^ *p* < 0.01 compared among the test groups. BMM, bone marrow macrophages; DC, dental calculus; M-CSF, macrophage colony-stimulating factor; NLRP3^−/−^-S, culture supernatant of the NLRP3-deficient mouse macrophages stimulated with DC; WT, wild-type; WT-S, culture supernatant of the wild-type mouse macrophages stimulated with DC.

**Figure 2 ijms-22-12434-f002:**
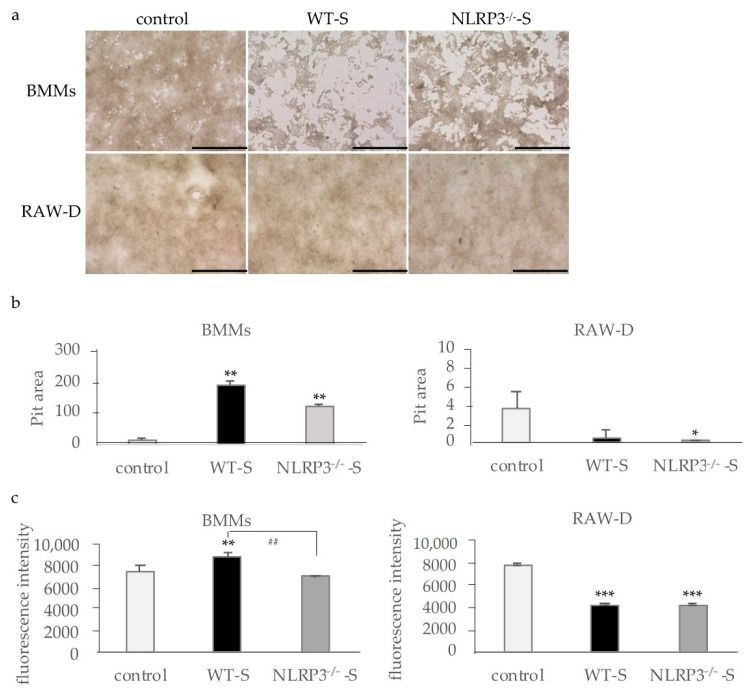
Evaluation of osteoclast bone resorption capacity. BMMs were incubated with 30 ng/mL M-CSF and 20 ng/mL RANKL for 48 h. Then, the cells were incubated for 72 h with the same concentrations of M-CSF, RANKL, and the culture supernatant of WT or NLRP3^−/−^ mouse macrophages stimulated with DC. RAW-D cells were incubated with 20 ng/mL RANKL for 48 h. Then, the cells were incubated for 72 h with the same concentrations of RANKL and the culture supernatant of WT or NLRP3^−/−^ mouse macrophages stimulated with DC. After removing the supernatants, the calcium-coated plate was treated with 5% sodium hypochlorite to remove the cells (**a**), and the pit area was measured with ImageJ (**b**). The bone resorption capacity was also analyzed by measuring the fluorescence intensity of the supernatants (**c**). Scale bar = 500 μm. The differences between the groups were analyzed by one-way ANOVA followed by a Tukey test for multiple comparisons. * *p* < 0.05 ** *p* < 0.01 *** *p* < 0.001 compared with the control. ^##^ *p* < 0.01 compared among the test groups. BMM, bone marrow macrophage; DC, dental calculus; NLRP3^−/−^-S, culture supernatant of the NLRP3-deficient mouse macrophages stimulated with DC; WT, wild-type; WT-S, culture supernatant of the wild-type mouse macrophages stimulated with DC.

**Figure 3 ijms-22-12434-f003:**
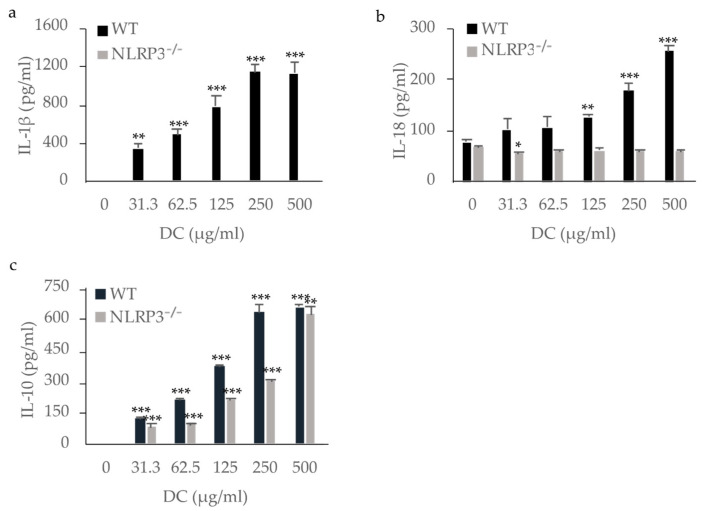
Production of IL-1β, IL-18, and IL-10 by mouse macrophages stimulated with dental calculus (DC). WT and NLRP3^−/−^ mouse macrophages were stimulated with DC for 8 h. The production of IL-1β (**a**), IL-18 (**b**), and IL-10 (**c**) were measured by ELISA. The differences between the groups were analyzed by one-way ANOVA followed by a Tukey test for multiple comparisons. * *p* < 0.05 ** *p* < 0.01 *** *p* < 0.001 compared with the unstimulated control. WT, wild-type.

**Figure 4 ijms-22-12434-f004:**
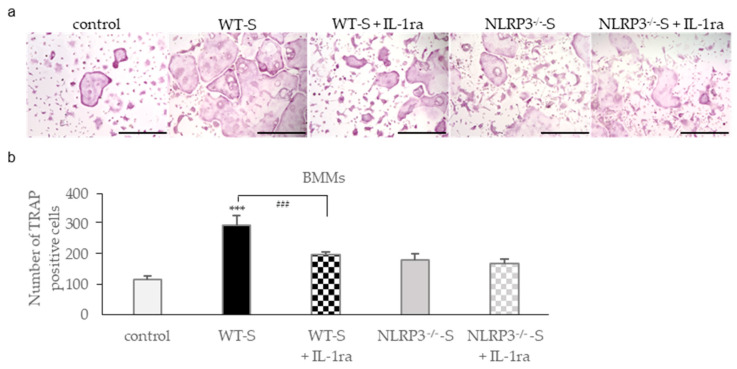
Effects of recombinant (r)IL-1ra on osteoclastogenesis. BMMs were incubated with 30 ng/mL M-CSF and 20 ng/mL RANKL for 48 h. Then, the cells were incubated for 24 h with the same concentration of M-CSF, RANKL, and culture supernatant from WT or NLRP3^−/−^ mouse macrophages stimulated with DC in the presence or absence of 10 μg/mL rIL-1ra. These cells were subjected to TRAP staining (**a**), and TRAP-positive cells with more than three nuclei were counted (**b**). Scale bar = 500 μm. The differences between the groups were analyzed by one-way ANOVA followed by a Tukey test for multiple comparisons. *** *p* < 0.001 compared with the control. ^###^
*p* < 0.001 compared among the test groups. BMM, bone marrow macrophage; M-CSF, macrophage colony-stimulating factor; NLRP3^−/−^-S, culture supernatant from NLRP3-deficient mouse macrophages stimulated with DC; TRAP, tartrate-resistant acid phosphatase; WT, wild-type; WT-S, culture supernatant from wild-type mouse macrophages stimulated with DC.

**Figure 5 ijms-22-12434-f005:**
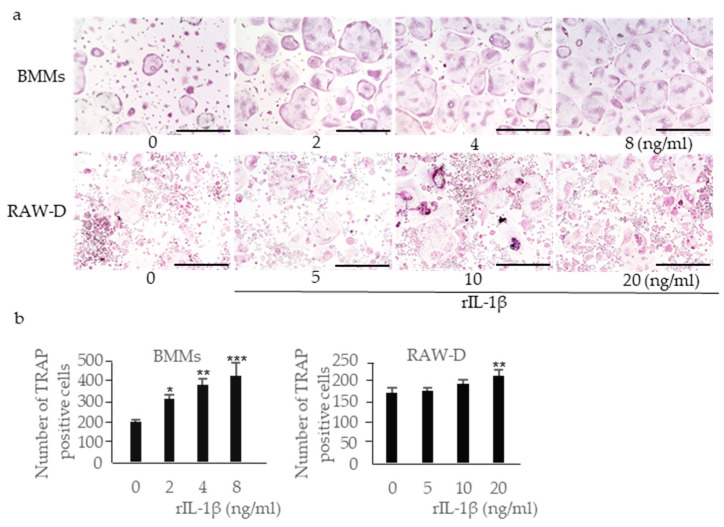
Effects of recombinant (r)IL-1β on osteoclastogenesis. BMMs were incubated with 30 ng/mL M-CSF and 20 ng/mL RANKL for 48 h. Then, the cells were incubated for 24 h with the same concentration of M-CSF, RANKL, and rIL-1β. RAW-D cells were incubated with 20 ng/mL RANKL for 48 h. Then, the cells were incubated for 48 h with the same concentration of RANKL and rIL-1β. These cells were subjected to TRAP staining (**a**), and TRAP-positive cells with more than three nuclei were counted (**b**). Scale bar = 500 μm. The differences between the groups were analyzed by one-way ANOVA followed by a Tukey test for multiple comparisons. * *p* < 0.05 ** *p* < 0.01 *** *p* < 0.001 compared with controls containing no rIL-1β. BMM, bone marrow macrophage; M-CSF, macrophage colony-stimulating factor; TRAP, tartrate-resistant acid phosphatase.

**Figure 6 ijms-22-12434-f006:**
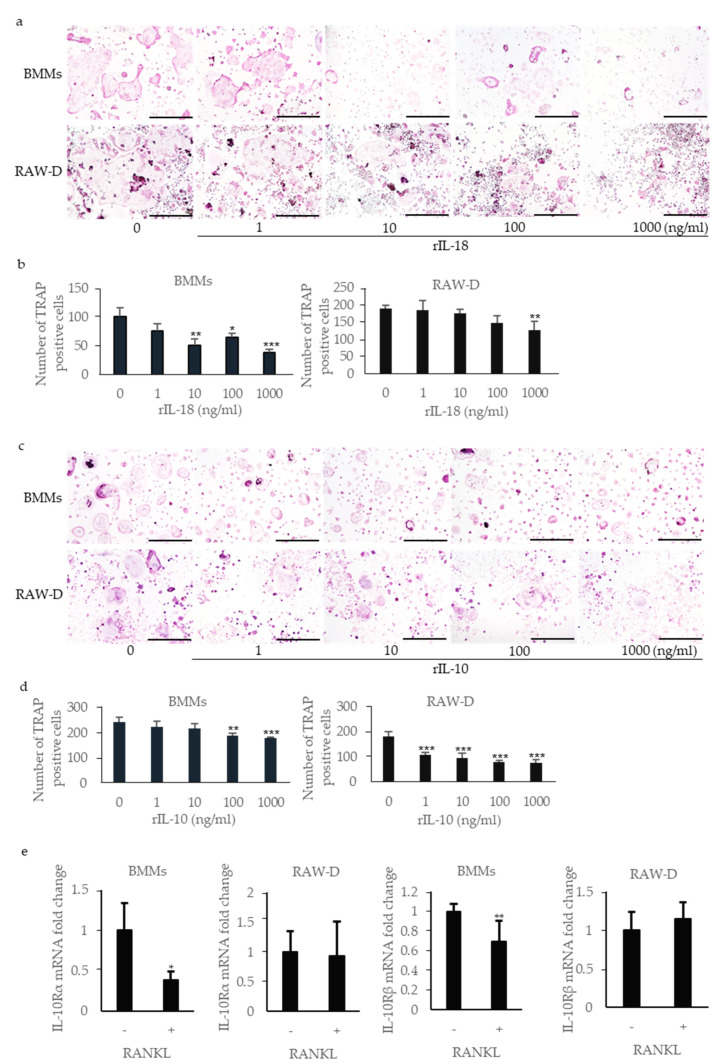
Effects of recombinant (r)IL-18 and rIL-10 on osteoclastogenesis. BMMs were incubated with 30 ng/mL M-CSF and 20 ng/mL RANKL for 48 h. Then, the cells were incubated for 24 h with the same concentration of M-CSF, RANKL, and rIL-18 (**a**,**b**) or rIL-10(**c**,**d**). RAW-D cells were incubated with 20 ng/mL RANKL for 48 h. Then, the cells were incubated for 48 h with the same concentration of RANKL and rIL-18 or rIL-10. These cells were subjected to TRAP staining (**a**,**c**), and TRAP-positive cells with more than three nuclei were counted (**b**,**d**). Total RNA was extracted from the unstimulated and RANK-primed cells, and the relative expression of *IL-10RA* and *IL-10RB* mRNA was determined using quantitative reverse transcription–polymerase chain reaction (qRT–PCR) (**e**). Scale bar = 500 μm. The differences between the groups were analyzed by one-way ANOVA followed by a Tukey test for multiple comparisons. The differences between two groups were analyzed using *t*-tests. * *p* < 0.05 ** *p* < 0.01 *** *p* < 0.001 compared with the control. BMM, bone marrow macrophage; M-CSF, macrophage colony-stimulating factor; TRAP, tartrate-resistant acid phosphatase.

**Figure 7 ijms-22-12434-f007:**
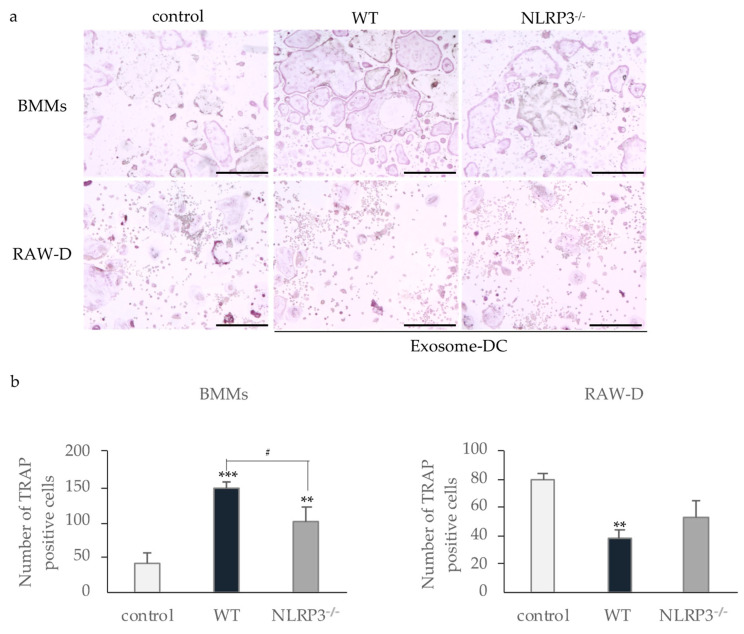
Effect of exosomes from mouse macrophages stimulated with DC. Exosomes were isolated from the culture supernatant of WT or NLRP3^−/−^ mouse macrophages stimulated with DC. BMMs were incubated with 30 ng/mL M-CSF and 20 ng/mL RANKL for 48 h. Then, the cells were incubated for 24 h with the same concentration of M-CSF, RANKL, and exosomes from WT or NLRP3^−/−^ mouse macrophages stimulated with DC. RAW-D cells were incubated with 20 ng/mL RANKL for 48 h. Then, the cells were incubated for 48 h with the same concentration of RANKL and exosomes from WT or NLRP3^−/−^ mouse macrophages stimulated with DC. These cells were subjected to TRAP staining (**a**), and TRAP-positive cells with more than three nuclei were counted (**b**). Scale bar = 250 μm. The differences between the groups were analyzed by one-way ANOVA followed by a Tukey test for multiple comparisons. ** *p* < 0.01 *** *p* < 0.001 compared with the control. ^#^ *p* < 0.05 compared among the test groups. BMM, bone marrow macrophage; DC, dental calculus; exosome-DC, exosome from culture supernatant of the mouse macrophages stimulated with DC; M-CSF, macrophage colony-stimulating factor; TRAP, tartrate-resistant acid phosphatase; WT, wild-type.

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
