# Peer review of "The Role of Cytokines Produced via the NLRP3 Inflammasome in Mouse Macrophages Stimulated with Dental Calculus in Osteoclastogenesis"

_ijms, 2021, doi:10.3390/ijms222212434_

Round 1

Reviewer 1 Report

Mae and colleagues present their work on the role of cytokines produced via the inflammasome on after stimulation by dental calculus (DC) on osteoclastogenesis. Although the aim of the paper was very promising and most of the data are convincing, there are some missing data and some strange data that don’t make this paper so strong and as it is right now probably not fit for publication.

The purpose of the paper is to investigate the role of the cytokines produced by mouse macrophages stimulated with DC via the NLRP3 inflamasome during osteoclastogenesis.

Maybe the data should be presented in another manner. The effect of supernatants from mouse macrophages stimulated with DC on osteoclast formation first, then the identification of some cytokines present in the supernatants. It would be stronger this way. By the way, only already osteoclastogenesis regulating cytokines have been studied. It would have been far more interesting to identify more cytokines secreted by macrophages after DC stimulation.

Why did the authors used RAW cells ? All the discussion about the opposite effect observed on RAW is not so convincing.

In the Figure legends, the statistical test should be noted.

The statistical test used should be changed. As the data are shown in the figures, only an ANOVA should have been done potentially followed by a post hoc test.

2.2. section

Actually, the BMM used for the osteoclast differentiation assay and the pit assay are primed (both by RANKL and MCSF). This should be stated in the text, not only in the legend.

2.3. section

The fact that the ILRa didn’t inhibit totally the effect of DC in the WT-S condition should be mentioned.

2.4. section

The point of these experiments is not very clear. Already known effects of already known cytolkines on osteoclastogenesis don’t bring any news or data to the purpose of the study.

The figure 5 and 6 should be merged. Alternatively, if 2 figures are needed, at least organized them, one for the stimulation of osteoclastogenesis, on for the inhibition of it.

Discussion section

Using RAW cells to conclude that supernatant exerted anti osteoclastogenesis is a little bit extreme.

The authors conclude that DC may play a clinically important role in alveolar bone resorption (line 268). Although without testing the effect of DC directly on osteoclastogenesis, this can’t be stated. In addition, as inflammasome exists in osteoclasts, it would of interest to analyse the effect of DC directly on osteoclast.

Material and methods section

It is only when the materials and methods section is read that the fact that immortalized BMM from WT and NLRP3 deficient mice have been used. How were they immortalized ? This piece of information questions the control used when those immortalized BMM were used. What was it exactly ? It would probably would be needed to have unstimulated BMM WT and NLRP3 KO supernatant used.*

For the bone resorption assay, the author didn’t say the name of the manufacturer.

And for the exosome experiment, what type of BMM were used WT BMM from BALB/c mice or immortalized BMM ?

Author Response

Reply to Reviewer 1:

(Manuscript ID: IGMS-1445735)

Thank you for giving us the important comments and suggestion.

We have addressed your comments as follows:

  1. The purpose of the paper is to investigate the role of the cytokines produced by mouse macrophages stimulated with DC via the NLRP3 inflammasome during osteoclastogenesis. Maybe the data should be presented in another manner. The effect of supernatants from mouse macrophages stimulated with DC on osteoclast formation first, then the identification of some cytokines present in the supernatants. It would be stronger this way.

According to the suggestion, Figure 2 was changed to new Figure 1, Figure 3 was changed to new Figure 2 and Figure 1 was changed to the new Figure 3.

  1. By the way, only already osteoclastogenesis regulating cytokines have been studied. It would have been far more interesting to identify more cytokines secreted by macrophages after DC stimulation.

We agree with Reviewer 1. We tried to identify the cytokines that have the unknown role on the osteoclastogenesis. However, we could not detect IL-33, IL-37 or IL-38 in the culture supernatants of WT mouse macrophages stimulated with DC. Therefore, we could not show these results.

  1. Why did the authors used RAW cells? All the discussion about the opposite effect observed on RAW is not so convincing.

RAW osteoclasts have been extensively employed in studies of osteoclastogenesis for more than 20 years, and have homogenous nature of osteoclast precursor populations (devoid of osteoblast, stromal, lymphocytes, and other cell types). Therefore, this cell line is suitable for evaluation of the effects of cytokines on osteoclast precursors. Although the majority of BMMs is osteoclast precursor cells, BMMs contain small number of other cell types that may indirectly regulate osteoclastogenesis. We have added this explanation to Discussion.

  1. In the Figure legends, the statistical test should be noted.

According to the suggestion, we have added “The differences between the groups were analyzed by one-way ANOVA followed by a Tukey test for multiple comparisons” to Figure legends.

  1. The statistical test used should be changed. As the data are shown in the figures, only an ANOVA should have been done potentially followed by a post hoc test.

According to the suggestion, we have changed the statistical method to one-factor ANOVA followed with the Tukey–Kramer test for the comparisons among the groups. Because we have added the qRT-PCR results to the new Figure 6, the differences between two groups were analyzed using t-tests.

  1. (2.2. section) Actually, the BMM used for the osteoclast differentiation assay and the pit assay are primed (both by RANKL and MCSF). This should be stated in the text, not only in the legend.

The BMMs used for the osteoclast differentiation assay and the pit assay were primed with RANKL and M-CSF. We have added this to the Results.

  1. (2.3. section) The fact that the IL-1Ra didn’t inhibit totally the effect of DC in the WT-S condition should be mentioned.

We mentioned “rIL-1ra suppressed their numbers to the same level as those in RANKL-primed BMMs incubated with the culture supernatant from NLRP3-deficient mouse macrophages stimulated with DC” in the Results. Because TNF-α and other pro-inflammatory cytokines existed in the culture supernatants of both WT and NLRP3-deficient mouse macrophages stimulated with DC, IL-1ra did not inhibit totally the effect of the culture supernatant of WT mouse macrophages stimulated with DC.  

  1. (2.4. section) The point of these experiments is not very clear. Already known effects of already known cytokines on osteoclastogenesis don’t bring any news or data to the purpose of the study.

We observed the opposite effects of the culture supernatants of mouse macrophages on RAW-D cells and BMMs. The osteoclastogenic effect of IL-1β was stronger in BMMs than that in RAW-D cells and anti-osteoclastogenic effect of IL-10 was stronger in RAW-D cells than that in BMMs. We believe this is one of the reasons to explain the opposite effect of culture supernatants. This was mentioned in the Discussion section.

  1. The figure 5 and 6 should be merged. Alternatively, if 2 figures are needed, at least organized them, one for the stimulation of osteoclastogenesis, on for the inhibition of it.

According to the suggestion, we have shown the results of experiments using rIL-1β in the new Figure 5 and those using rIL-18 and rIL-10 in the new Figure 6. In addition, we have added the results of qRT-PCR to the new Figure 6.

  1. (Discussion section) Using RAW cells to conclude that supernatant exerted anti osteoclastogenesis is a little bit extreme.

According to the suggestion, we have replaced the phrase ‘anti-osteoclastogenic’ with ‘possible anti-osteoclastogenic’.

  1. The authors conclude that DC may play a clinically important role in alveolar bone resorption (line 268). Although without testing the effect of DC directly on osteoclastogenesis, this can’t be stated. In addition, as inflammasome exists in osteoclasts, it would of interest to analyse the effect of DC directly on osteoclast.

It is interesting to analyze the direct effect of DC on osteoclastogenesis. However, DC precipitates on the tooth surfaces and hardly penetrate the epithelium. Therefore, it is unlikely that DC directly stimulate the osteoclast precursors in the clinical situation. We explained it in the Discussion section.

  1. (Material and methods section) It is only when the materials and methods section is read that the fact that immortalized BMM from WT and NLRP3 deficient mice have been used. How were they immortalized?

Immortalized macrophage cell lines were generated with a J2 recombinant retrovirus (carrying v-myc and v-raf(mil) oncogenes). We have added this description to Materials and Methods section. In addition, we have added ‘immortalized wild-type (WT) and NLRP3-deficient mouse macrophages’ to the first appearance of the Results.

  1. This piece of information questions the control used when those immortalized BMM were used. What was it exactly ? It would probably would be needed to have unstimulated BMM WT and NLRP3 KO supernatant used.*

We incubated RANKL-primed BMMs and RAW-D cells with the culture supernatants of unstimulated WT or NLRP3-deficient mouse macrophages. These culture supernatants did not affect the osteoclastogenesis. We have shown these results in the Supplementary Figure 1 and added this description to Results.

  1. For the bone resorption assay, the author didn’t say the name of the manufacturer.

Bone resorption assay kit 48 was purchased from Iwai Chemicals Company (Tokyo, Japan). We mentioned it in 4.1. Reagents section in Materials and Methods.

  1. And for the exosome experiment, what type of BMM were used WT BMM from BALB/c mice or immortalized BMM ?

We isolated exosomes from the culture supernatants of immortalized mouse macrophages. We have added ‘immortalized’ to the first sentence of 4.8. Exosome isolation section in Material and Methods.

Reviewer 2 Report

Authors investigated the role of NLRP3-dependent and -independent cytokines from mouse macrophages on osteoclastogenesis and showed that the IL-1beta promotes osteoclastogenesis but IL-18 and IL-10 inhibit osteoclastogenesis. Also, authors showed there are some difference between cell types (BMMs vs RAW-D). In general, this manuscript was well written and authors provided adequate data to support the main conclusion. Specific comments are listed below.

1, Authors used the NLRP3-deficient macrophages for this study. From the therapeutic point of view, does supernatants from wild type macrophages treated with NLRP3 inflammasome inhibitors (e.g. mcc950, YVAD) have the similar results? authors could provide additional data for this or discuss this.

2. Since thee cytokines have different effects on osteoclastogenesis in cells, do these effects reflect the differential expression pattern of their receptors in cells? Western blot for the levels of receptors or RT-qPCR will clarify this.

3. Authors should elaborate a little more on how cytokines within Exosomes affect the osteoclastogenesis. Do these cytokines expose on the surface of exosomes?       

Author Response

Reply to Reviewer 2:

(Manuscript ID: IGMS-1445735)

Thank you for giving us the important comments and suggestion.

We have addressed your comments as follows:

  1. Authors used the NLRP3-deficient macrophages for this study. From the therapeutic point of view, does supernatants from wild type macrophages treated with NLRP3 inflammasome inhibitors (e.g. mcc950, YVAD) have the similar results? authors could provide additional data for this or discuss this.

As mentioned by Reviewer 2, we believe that the NLRP3 inflammasome inhibitors, such as MCC950 and z-YVAD-fmk, can inhibit the IL-1β production. From the therapeutic point of view, the inhibition of IL-1β production is important. We have added “From the clinical point of view, application of NLRP3 inflammasome inhibitors, such as MCC950 and glyburide, may be useful for prevention of this alveolar bone resorption.” to Discussion.

  1. Since thee cytokines have different effects on osteoclastogenesis in cells, do these effects reflect the differential expression pattern of their receptors in cells? Western blot for the levels of receptors or RT-qPCR will clarify this.

Because the inhibitory effect of rIL-10 in RAW-D cells was considerably stronger than that in RANKL-primed BMMs, we have analyzed the expression of the receptors for IL-10 by qRT-PCR. As shown in the new Figure 6, the expression of IL-10 receptors was downregulated by RANKL-priming in BMMs. The downregulation of IL-10 receptors may contribute to the low sensitivity of RANKL-primed BMMs to IL-10.

  1. Authors should elaborate a little more on how cytokines within Exosomes affect the osteoclastogenesis. Do these cytokines expose on the surface of exosomes?

Exosomes are approximately 100-nm secreted vesicles that contain many signaling molecules, such as microRNAs (miRNAs), messenger RNAs, and proteins. Exosomes released from the host cell surface can fuse with the plasma membranes of recipient cells and deliver their contents into the cytoplasm. The components delivered by exosomes to recipient cells result in the alteration of biological response. We have added these description to Discussion.